# The Genome of *Varunaivibrio sulfuroxidans* Strain TC8^T^, a Metabolically Versatile Alphaproteobacterium from the Tor Caldara Gas Vents in the Tyrrhenian Sea

**DOI:** 10.3390/microorganisms11061366

**Published:** 2023-05-23

**Authors:** Sushmita Patwardhan, Jonathan Phan, Francesco Smedile, Costantino Vetriani

**Affiliations:** 1Department of Marine and Coastal Sciences, Rutgers University, New Brunswick, NJ 08901, USA; 2Department of Biochemistry and Microbiology, Rutgers University, New Brunswick, NJ 08901, USA

**Keywords:** shallow-water vent, Tyrrhenian Sea, Tor Caldara, chemolithotroph, *Alphaproteobacteria*, *Thalassospiraceae*, genome, geothermal, sulfur oxidation, DNA uptake, nitrogen fixation, carbon fixation

## Abstract

Varunaivibrio *sulfuroxidans* type strain TC8^T^ is a mesophilic, facultatively anaerobic, facultatively chemolithoautotrophic alphaproteobacterium isolated from a sulfidic shallow-water marine gas vent located at Tor Caldara, Tyrrhenian Sea, Italy. *V. sulfuroxidans* belongs to the family *Thalassospiraceae* within the Alphaproteobacteria, with *Magnetovibrio blakemorei* as its closest relative. The genome of *V. sulfuroxidans* encodes the genes involved in sulfur, thiosulfate and sulfide oxidation, as well as nitrate and oxygen respiration. The genome encodes the genes involved in carbon fixation via the Calvin–Benson–Bassham cycle, in addition to genes involved in glycolysis and the TCA cycle, indicating a mixotrophic lifestyle. Genes involved in the detoxification of mercury and arsenate are also present. The genome also encodes a complete flagellar complex, one intact prophage and one CRISPR, as well as a putative DNA uptake mechanism mediated by the type IVc (aka Tad pilus) secretion system. Overall, the genome of *Varunaivibrio sulfuroxidans* highlights the organism’s metabolic versatility, a characteristic that makes this strain well-adapted to the dynamic environmental conditions of sulfidic gas vents.

## 1. Introduction

Hydrothermal and gas vents are manifestations of volcanism on Earth. Both environments are characterized by the venting of gas and/or hydrothermal fluids, enriched in reduced, inorganic chemical species into the oxygenated water column. This creates a redox disequilibrium, which is then harnessed by lithotrophic microorganisms that convert chemical energy into ATP. Tor Caldara is a shallow-water gas vent located in the Mediterranean Sea, near the town of Anzio, Italy. The composition of gases released by the Tor Caldara vents is mostly CO_2_ (avg. 77 mol%) and H_2_S (avg. 23 mol%) [1]. The Tor Caldara gas vents are generated by the degassing of CO_2_ of deep magmatic/mantle origin due to the alkali-potassic volcanoes present in the region, in particular the Colli Albani volcanic complex [2]. Since the gas venting at Tor Caldara is a peripheral manifestation of the quiescent Colli Albani volcano, whose activity began 600 thousand years ago [2], the system is thought to be very stable over time in comparison to the more dynamic venting associated with active hydrothermal vents. 

The continuous venting of CO_2_ and H_2_S gas from the seafloor at Tor Caldara supports substrate-attached chemosynthetic microbial biofilms mostly composed of members of the *Camplylobacterota* (aka *Epsilonproteobacteria*) and *Gammaproteobacteria* [1,3], as well as more diverse microbial communities associated with the vent-associated sediments [4]; this is in line with previous studies of coastal and deep-sea marine hydrothermal environments (e.g., [5,6,7,8,9,10]). *Alphaproteobacteria* have also been observed to contribute abundantly to shallow-water vent microbial communities [11,12,13,14]. Consistent with these observations, *Alphaproteobacteria* were relatively abundant at Tor Caldara (4.5 to 15.1% of the vent-associated sediment microbial community and 6 to 10% of the substrate-attached biofilms) [1,4], and a novel genus of sulfur-oxidizing, chemolithoautotrophic alphaproteobacterium, *Varunaivibrio sulfuroxidans* strain TC8^T^, was isolated and characterized from the sediments associated with the gas vents of this site [15]. Currently, *V. sulfuroxidans* belongs to the family *Thalassospiraceae* (although a recent study proposed its reassignment to the new family *Magnetovibrionaceae*) [16], which includes four genera of marine bacteria: *Magnetospira*, *Magnetovibrio*, *Thalassospira* and *Varunaivibrio* [17]. In particular, *Magnetospira thiophila* strain MMS-1^T^ and *Magnetovibrio blakemorei* strain MV-1^T^ are magnetotactic bacteria isolated from salt marshes [18,19]. *M thiophila*, *M. blakemorei* and *Varunaivibrio sulfuroxidans* are facultative chemolithoautotrophs that conserve energy by anaerobic (nitrate) and/or microaerobic respiration, with reduced sulfur species as electron donors [15,18,19]. 

In this study, we obtained the genome sequence of *V. sulfuroxidans* strain TC8^T^ and we analyzed it in light of its known physiological and metabolic traits, we reconstructed its central metabolism, its heavy metal detoxification and its DNA uptake mechanism. This is one of the few chemolithoautotrophic alphaproteobacterial genomes that have been sequenced from a shallow-water hydrothermal vent, another being a *Rhodovulum* sp. from the coastal hydrothermal system of Kueishantao Island, Taiwan [20].

## 2. Materials and Methods

### 2.1. Genome Project History

The genome of *V. sulfuroxidans* strain TC8^T^ was sequenced in August 2016 as part of a genome sequencing effort of newly characterized bacterial strains isolated at the Deep-Sea Microbiology Lab at Rutgers University (http://marine.rutgers.edu/deep-seamicrobiology/, accessed on 3 February 2016). The draft genome of *V. sulfuroxidans* was sequenced at the MicrobesNG facility (http://microbesng.uk/, accessed on 14 June 2016), and it was closed in-house using the Oxford Nanopore MinION platform [21].

### 2.2. Growth Conditions and Genomic DNA Preparation

*Varunaivibrio sulfuroxidans* strain TC8^T^ was grown in modified medium 1011 [22] supplemented with 0.2% (*w*/*v*) KNO_3_ under a gas phase of N_2_/CO_2_ (80:20 *v*/*v*; 200 kPA) according to Patwardhan and Vetriani, 2016 [15]. The genomic DNA was extracted from 0.5–1 g of cell biomass following a phenol:chloroform extraction protocol. In short, the bacterial pellet was added to 850 μL extraction buffer (50 mM Tris-HCl, 20 mM EDTA, 100 mM NaCl; pH 8.0) supplemented with 100 μL of lysozyme (100 mg/mL) and incubated at 37 °C for 30 min. This mix was then supplemented with 5 μL of proteinase K (20 mg/mL) and incubated at 37 °C for 30 min followed by the addition of 50 μL SDS (20%) and further incubated at 65 °C for 1 h. The genomic DNA was extracted by performing two consecutive phenol:chloroform:isoamylalcohol (25:24:1) extractions. The supernatant was precipitated in 3 M sodium-acetate and isopropanol, washed twice with 70% ice-cold ethanol and re-suspended in ultra-pure water.

### 2.3. Genome Sequencing and Assembly

A hybrid approach involving short Illumina reads and long Nanopore MinIon reads was used to sequence the genome of *V. sulfuroxidans*. Genomic DNA was sequenced using a paired end 2 × 250 bp MiSeq library which generated a total of 1,004,710 quality-checked short reads. A total of 40,150 long reads were also obtained on the Naonopre MinIon platform using the SQK-RAD004 library prep kit. Obtained short and long reads were assembled successively using Unicycler [23] and Geneious [24]. Assembly yielded 1 complete contig of a total length of 3,066,297 bp, with a 138× sequencing coverage. The genome sequencing project of *V. sulfuroxidans* strain TC8^T^ is available under accession number CP119676.

### 2.4. Genome Annotation

The genome assembly was annotated using the Joint Genome Institute IMG system [25] and the RAST (Rapid Annotation using Subsystem Technology) server. The predicted proteins of *Varunaivibrio sulfuroxidans* TC8^T^ involved in key metabolic pathways were searched using the KEGG database [26]. CRISPERs and tandem repeats were identified using CRISPRFinder [27]. Prophages in the genome were searched using Phaster [28]. Genomic islands were identified using IslandViewer [29]. The online software Alienness Web Server V.2.0 was used to detect evidence of possible horizontal transfer events [30].

### 2.5. Phylogenetic, Average Nucleotide Identity and Digital DNA–DNA Hybridization Analyses

Phylogenetic analyses were carried out as previously described [8,31]. In brief, 16S rRNA gene sequences were aligned with ClustalO [32], and the alignment was manually refined using SEAVIEW version 5.0.5 [33]. The maximum likelihood phylogeny tree was inferred from the alignment using PhyML version 3.1 [34], with the Jukes and Cantor substitution model [35] and 500 bootstrap resamplings. The amino acid sequences of NifH, ComEC and CpaC from strain TC-8^T^ and closest homologs were retrieved from GenBank and aligned in SEAVIEW [33], and the maximum likelihood phylogeny trees were inferred from the alignment using PhyML [34], with the Li and Gasquel (LG) aminoacid replacement matrix [36] and 500 bootstrap resamplings. The average nucleotide identity (ANI) between strain TC8^T^ and its closest relative, *Magnetovibrio blakemorei*, was calculated using the EZBioCloud tool [37]. The digital DNA–DNA hybridization (dDDH) value between strain TC8^T^ and *M. blakemorei* was obtained using the genome-to-genome distance calculator (GGDC) [38].

## 3. Results and Discussion

### 3.1. Overview of Varunaivibrio Sulfuroxidans Strain TC8^T^

*V. sulfuroxidans* strain TC8^T^ is a mesophilic, facultatively anaerobic, facultatively chemolithoautotrophic bacterium isolated from a sulfidic shallow-water marine gas vent located at Tor Caldara in the Tyrrhenian Sea, Italy [15]. Cells are Gram-stain-negative curved rods with one or more polar flagella. Cells were approximately 1–1.5 µm in length and 0.6 µm in width. Strain TC8^T^ grows optimally at 30 °C, with 15–20 g NaCl l/L and an optimum pH between 6.0 and 7.0. The generation time under optimal conditions is 8 h. Strain TC8^T^ is a facultative chemolithoautotroph also capable of using organic substrates (tryptone, peptone, Casamino acids, pyruvate and glycerol). Chemolithoautotrophic growth occurs with sulfur and thiosulfate as the electron donors, CO_2_ as the carbon source and nitrate or oxygen (5 %, *v*/*v*) as the electron acceptors. Strain TC8^T^ is able to grow in the absence of a source of fixed nitrogen (nitrate or ammonium) under a headspace of N_2_/CO_2_/O_2_ (53:45:2 *v*/*v*), which implies that it is able to fix gaseous nitrogen. 

Phylogenetic analysis of the 16S rRNA gene sequence of strain TC8^T^ placed this organism in a lineage within the family *Thalassospiraceae*, with *Magnetovibrio blakemorei* strain MV1^T^ (91.25% sequence identity) as its closest cultured relative (Figure 1). When uncultured microorganisms known solely by their 16S rRNA gene are included in the analysis, the closest sequence to *V. sulfuroxidans* is clone 7M24_051 (98.89% sequence identity; Figure 1), which is obtained from an iron-rich, inactive sulfide associated with the deep-sea hydrothermal vent system of the East Pacific Rise at 9° north [39]. 

While both *M. blakemorei* strain MV1^T^ and *M. thiophila* strain MMS-1^T^ are magnetotactic, *V. sulfuroxidans* strain TC8^T^ is not; formation of magnetosomes under optimal anaerobic conditions was not observed in transmission electron micrographs of thin sections of cells, and the *mam* genes are not present in its genome (Appendix A) [15].

### 3.2. Genome Structure

The genome of *V. sulfuroxidans* strain TC8^T^ contains 3,066,297 bp, with coding regions taking 88.1% of the genome and a G+C content of 54.4% (Table 1). The most abundant gene categories are assigned to the clusters of orthologous genes (COGs) database code for amino acid transport and metabolism (10.3%), energy conversion (8.9%), translation and ribosomal structure (7.6%) and inorganic ion transport and metabolism (7.5%; Table 2). Furthermore, 27.1% of the genes of strain TC8^T^ were not in the COG database, while 4.7% were classified as unknown function (Table 2). 

The genome structure of *V. sulfuroxidans* strain TC8^T^, including the gene distribution, predicted genomic islands, prophage and genes involved in membrane transport, carbon and energy metabolism is summarized in Figure 2. Nine regions of the genome of strain TC8^T^ were predicted as possible genomic islands (Figure 2A, line 2), while one region was found to be a complete bacteriophage (Figure 2B). The intact prophage found in the genome of TC8^T^ has a size of 33.4Kb, with a total of 48 proteins and a GC content of 65.53% (Figure 2B). It consists of 22 different phage-like proteins, with the majority (31%) of proteins similar to prophage vB_RhkS_P1, which was induced in *Rhodovulum* sp. strain P5, an alphaproteobacterium isolated from a shallow-water hydrothermal vent off Kueishantao Island [20,40].

The genome of strain TC8^T^ encodes one short CRISPR array, which is 431 bp in length and has six spacers. The spacers did not show sufficient alignment to any sequences in the Genbank Phage or RefSeq plasmid databases. Our analysis of the genome structure of strain TC8^T^ did not reveal the presence of plasmids.

The average nucleotide identity (ANI) between the genomes of *V. sulfuroxidans* and its closest cultured relative, *M. blakemorei,* is 68.86%, while the estimated digital DNA–DNA hybridization (dDDH) between the two genomes is 18.7%, confirming their assignment to different genera. 

### 3.3. Carbon Metabolism and Microaerobic Respiration

The genome of strain TC8^T^ encodes for all the enzymes necessary for carbon fixation using the Calvin–Benson–Bassham (CBB) cycle, including the key enzymes ribulose 1,5-bisphosphate carboxylase (RubisCO) and phosphoribulokinase (Figure 2A and Figure 3 and Appendix A). The RubisCO enzyme catalyzes the carboxylation of ribulose 1,5-bisphosphate, which results in two molecules of 3-phosphoglycerate. Presently, there are three forms of RubisCO known that catalyze the CO_2_ fixation reaction, with form I occurring the most dominantly [41]. Form II, encoded by the *cbbM* gene, was first isolated from the alphaproteobacterium, *Rhodospirillum rubrum.* Similarly to other members of the *Thalassospiraceae* family, the genome of *V. sulfuroxidans* TC8^T^ encodes a form II RubisCO closely related to the enzyme from *M. blakemorei* (sequence identity 88.45%). The carbon-concentrating mechanisms of strain TC8^T^ include a type γ carbonic anhydrase, which catalyzes the interconversion between the dissociated ions of carbonic acid and carbon dioxide and water (Figure 3 and Appendix A). The amino acid sequence of the carbonic anhydrase of *V. sulfuroxidans* is 69.4% identical to that of *M. blakemorei*.

The genome of strain TC8^T^ also encodes all the enzymes involved in the breakdown of organic carbon via the glycolysis pathway as well as the enzymes involved in the tricarboxylic acid (TCA) cycle, which provides reducing power for oxidative phosphorylation in heterotrophs (Figure 2 and Appendix A). Thus, the genome of *V. sulfuroxidans* strain TC8^T^ encodes for a heterotrophic as well as autotrophic metabolism, which is in line with its experimentally determined metabolic characteristics [15].

The genome of strain TC8^T^ encodes the entire microaerobic respiratory chain, including the high-affinity cbb3 type cytochrome oxidase and cytochrome c oxidase, consistent with the experimental evidence for microaerophilic growth (Figure 3) [15].

### 3.4. Sulfur Metabolism 

*V. sulfuroxidans* TC8^T^ was originally described as a sulfur-oxidizing chemolithoautotroph [15]. Accordingly, the genome encodes the SoxABXYZ pathway that is responsible for thiosulfate oxidation (Figure 3 and Appendix A). Strain TC8^T^ lacks the SoxCD component which, in *Paracoccus pantotrophus,* is designated as a sulfur dehydrogenase [42]. Without SoxCD_,_ thiosulfate is likely oxidized to elemental sulfur and not all the way to sulfate, as demonstrated experimentally in *Allochromatium vinosum* [43]. The aminoacid sequence of *V. sulfuroxidans* TC8^T^ SoxA is 65.83% identical to that of the *M. blakemorei* enzyme. Besides the Sox gene cluster, the genome of strain TC8^T^ encodes the sulfide:quinone oxidoreductases (SQR, 79.6% identical to the *M. blakemorei* enzyme) and a putative flavocytochrome c-sulfide dehydrogenase (Fcc; Figure 3 and Appendix A). The presence of these enzymes suggests that strain TC8^T^ has the potential to oxidize sulfide to elemental sulfur [44]. Current knowledge indicates that, in chemolithotrophic sulfide-oxidizing bacteria, SQR-mediated sulfide oxidation is more energetically favorable than sulfide oxidation by Fcc, and that overall SQR is a more efficient enzyme [45]. However, due to its high affinity for sulfide, Fcc is hypothesized to be selectively expressed at low sulfide concentrations [3,46]. The *dsr* gene cluster, encoding several enzymes involved in the oxidation of stored sulfur, is also present in the genome of strain TC8^T^ (Figure 3 and Appendix A). According to the current understanding, sulfur is brought into the cytoplasm in a persulfidic form (e.g., glutathione persulfide) [44]. DsrL releases the sulfide from the persulfide, which then passes on to the reversely operating sulfite reductase DsrAB via DsrC and DsrEFH [47]. The DsrAB oxidizes sulfide to sulfite. Next, DsrAB interacts with the trans-membrane DsrMKJOP complex, which passes electrons into the respiratory electron chain. Genes encoding for all the aforementioned proteins are present in the genome of strain TC8^T^. The sulfite formed is likely further oxidized to sulfate via AprAB and SAT enzymes, both encoded by the genome of strain TC8^T^. The presence of several sulfur oxidation pathways in the genome of strain TC8^T^ indicates its metabolic flexibility and its ability to oxidize a variety of reduced sulfur compounds to conserve energy, which likely provides this bacterium with an advantage to thrive in its sulfidic habitat.

### 3.5. Nitrogen Metabolism

Dissimilatory nitrate reduction to ammonia (DNRA) and denitrification are two key pathways involved in nitrogen metabolism that occur commonly in vent organisms [48,49]. *V. sulfuroxidans* conserves energy via the oxidation of reduced sulfur species coupled with the reduction of nitrate to dinitrogen gas [15]. In line with this experimental evidence, the genome *V. sulfuroxidans* encodes all four proteins involved in the complete denitrification pathway (Figure 3 and Appendix A). The first step of this pathway, i.e., the reduction of nitrate to nitrite, is catalyzed by a nitrate reductase, either the membrane bound NarG or the periplasmic NapAB. All four genes encoding the membrane bound nitrate reductase complex, namely, *narG*, *narH*, *narJ* and *narI,* are present in the genome of strain TC8^T^. The *napA* and *napB* genes encoding the periplasmic nitrate reductase NapAB are also present [48]. The aminoacid sequence of the periplasmic nitrate reductase (NapA) of *V. sulfuroxidans* is 80.26% identical to that of *M. blakemorei*. The genome has the entire *nap* gene cluster consisting of the quinol dehydrogenase NapC, the maturation chaperone NapD, the membrane-bound dehydrogenases NapG and NapH and the subunit NapF, whose role remains unclear (Figure 3). Several microorganisms, including *Escherichia coli*, encode both the membrane-bound (Nar) and periplasmic (Nap) nitrate reductases. Potter et al. (1999) experimentally demonstrated that, in nitrate-limited conditions, the presence of only the high-affinity Nap enzyme afforded *E. coli* a selective advantage over mutant strains that only encoded the Nar enzyme [50]. We hypothesize that, in the dynamic habitat of the Tor Caldara gas vents, the expression of the two nitrate reductases in *V. sulfuroxidans* is modulated by the availability of nitrate. The reduction of nitrite to nitric oxide (NO) is carried out by cytochrome cd1 protein NirS, which is encoded by the *nirS* gene. NorBC further reduces two molecules of NO to nitrous oxide (N_2_O). NosZ is involved in the reduction of N_2_O to N_2_, albeit it is not present in all organisms [51]. However, NosZ is present in the genome of strain TC8^T^, which is consistent with its ability to generate dinitrogen gas during growth under nitrate-reducing conditions [15]. Along with dissimilatory nitrate reduction, the genome also shows potential for assimilatory nitrate reduction, as the *nasA* and *nasB* genes that encode for cytoplasmic nitrate reductase NasAB are present (Figure 3). Genes for the assimilatory nitrite reductases NIT-6 and NirA were absent. However, the genome encoded for the NirBD protein and has also been shown to be involved in the assimilatory nitrite reduction [52,53].

Nitrogen fixation, namely, the reduction of N_2_ gas to two molecules of ammonia, is an important part of the marine nitrogen cycle, since it provides a source of newly fixed nitrogen to the ecosystem. The nitrogenase complex catalyzing this reaction is made up of two proteins: dinitrogenase reductase and dinitrogenase encoded by three genes, *nifH*, *nifD* and *nifK* [54]. All three genes are found in the genome of *V. sulfuroxidans*, which is consistent with its ability to grow in the absence of fixed nitrogen (Figure 3 and Appendix A) [15]. The genome of strain TC8^T^ also encodes a nitrogenase-associated Fe-S ferredoxin (*fdx* gene; Appendix A), which is 44.7% identical to that of *Rhodobacter capsulatus*. We hypothesize that the *fdx*-encoded ferredoxin serves as an electron donor to nitrogenase in *V. sulfuroxidans*, as experimentally demonstrated for *R. capsulatus* [55]. Phylogenetic analysis of the NifH placed the enzyme of *V. sulfuroxidans* in a cluster that includes several members of marine Alphaproteobacteria, including *M. blakemorei*, *Cohaesibacter marisflavi*, *C. intestine* and *Rhodobium orientis*, as well as *Martelella endophytica*, which was isolated from the rose rhizosphere (Figure 4). The NifH aminoacid sequences of *V. sulfuroxidans* and *M. blakemorei* are 88.8% identical. The genome of *V. sulfuroxidans* encodes several enzymatic complexes that catalyze the oxidoreduction of nitrogen (nitrate reduction, assimilation and nitrogen fixation) and sulfur species (thiosulfate, sulfur and possibly sulfide oxidation), effectively linking the nitrogen and sulfur cycles.

### 3.6. Heavy Metal Detoxification

Hydrothermal vent environments are known to have elevated concentrations of toxic heavy metals; hence, the resident microorganisms encode genes involved in heavy metal detoxification [56,57,58,59]. The genome of strain TC8^T^ contains both the *ars* and the *mer* operons involved in the detoxification of arsenate and mercury, respectively (Figure 3 and Appendix A). Arsenic in the form of arsenite or arsenate is toxic for most microorganisms, and resistance is associated with a three-gene or five-gene *ars* operon. The five-gene *ars* operon in strain TC8^T^ consists of arsenate reductase ArsC, membrane efflux protein ArsB, two trans-acting repressors ArsR and ArsD, as well as another efflux protein ArsB that infers higher resistance to arsenic [60,61]. The *mer* operon of strain TC8^T^ encodes for the periplasmic mercury-binding protein MerP, the trans-membrane transport protein MerT and the mercury reductase MerA [62,63]. *V. sulfuroxidans* did not grow in the presence of arsenate as the electron acceptor [15]. The genome does not encode for a selenate reductase; however, it has been shown that bacterial nitrate reductases, which the genome encodes for, are also capable of reducing selenate [64,65]. The genome also encodes for the DedA protein, which has been shown to be involved in selenite resistance [66]. The resistance of *V. sulfuroxidans* to mercury and selenite was not tested.

### 3.7. DNA Uptake System

Knowledge of extracellular DNA uptake in *Alphaproteobacteria* is limited; however, recent studies suggest the involvement of a type IVc pilus known as the Tad pilus system [67]. In the naturally competent plant pathogen, *Liberibacter crescens* strain BT1, the Tad pilus aids in survival through surface attachment and DNA uptake for nucleotide metabolism [68,69]. *L. crescens,* a member of the family *Rhizobiaceae* within the Alphaproteobacteria, was experimentally proven to be naturally competent, and the genes involved in this system were identified and found to be conserved among *Liberibacter* spp. [67,69]. 

Therefore, we used *L. crescens* strain BT1 as a reference organism to identify the putative DNA uptake system of *V. sulfuroxidans*. We discovered that the genome of *V. sulfuroxidans* encodes genes homologous to the *tad* genes organized in a cluster similar in structure to that of *M. blakemorei* and *L. crescens* (Figure 5). The Tad pilus system consists of a biogenesis ATPase (tadZ/cpaE), motor ATPase (tadA/cpaF), prepilin peptidase (tadV/cpaA), inner membrane staging complex (tadB/cpaG and tadC/cpaH), secretin (rcpA/cpaC), periplasmatic subunits (tadG/cpaB and rcpB/cpaD), inner membrane anchor (rcpC/tadG) and pilotin (tadD/cpaO) [67]. *V. sulfuroxidans* have homologs to genes in this system except for pilotin (Figure 3). The genome of strain TC8^T^ also encodes the DNA translocation machinery homologous to *L. crescens* strain BT1, as well as core genes involved in DNA uptake homologous to those encoded by *L. crescens*. Phylogenetic tree reconstruction of CpaC, the enzyme secretin of the type IVc secretion system, revealed that the enzyme of *V. sulfuroxidans* was most closely related to that of *M. blakemorei* (44.79% aminoacid sequence identity; Figure 6). The CpaC from *L. crescens* was placed in a cluster with homologous enzymes from other members of the family *Rhizobiaceae* (33.48% aminoacid sequence identity with the CpaC from *V. sulfuroxidans*).

ComEC is also necessary for natural competence and DNA internalization in *L. crescens*, and a homolog was found in strain TC8^T^ [69]. Phylogenetic analysis showed that the competence protein ComEC of *V. sulfuroxidans* was related to the homologous protein from *M. blakemorei* (45.47% aminoacid identity; Figure 7), while ComEC from *L. crescens* clustered with a group of proteins from various strains of *Rhizobium* and was 32.3% identical to that of *V. sulfuroxidans.*

Based on these results, we hypothesize that *V. sulfuroxidians* and *M. blakemorei* are naturally competent via the Tad pilus/Com system. However, the use of the Tad pilus system might differ between *L. crescens* and *V. sulfuroxidans* in the fact that former depends on this system for the essential uptake of nucleotides while strain TC8^T^ can synthesize its own nucleotides.

## 4. Conclusions

In this study, we sequenced and analyzed the genome of *V. sulfuroxidans* strain TC8^T^, whose physiology and metabolism were previously experimentally characterized. The genome of strain TC8^T^ revealed that this facultative chemolithoautotroph fixes carbon via the CBB cycle, oxidizes reduced sulfur species via the Sox, SQR and the oxidative Dsr pathways, encodes the nitrogenase gene cluster for nitrogen fixation and reduces nitrate via the periplasmic and/or cytoplasmic nitrate reductase complexes, Nap and Nar. Based on homology with the experimentally verified DNA uptake system of *L. crescens*, we further identified the Tad-mediated secretion system and competence gene cluster in the genome of *V. sulfuroxidans* and its closest relative, *M. blakemorei*. These findings suggest that both *V. sulfuroxidans* and *M. blakemorei* are naturally competent and have the genetic machinery to take up exogenous DNA. However, we did not find evidence of horizontally transferred genes in the genome of *V. sulfuroxidans*.

Linking the genome sequence of *V. sulfuroxidans* with its observed metabolism establishes a direct association between genotype and phenotype and enables genome-based metabolic inferences from environmental, sequence-based surveys, e.g., metagenome-assembled genomes (MAGs) of closely related strains (e.g., close relatives of *V. sulfuroxidans* identified by sequencing in hydrothermal and sedimentary habitats).

Overall, the genome of *V. sulfuroxidans*, along with its experimentally validated phenotype, reveals that this bacterium is metabolically versatile and corroborates its ability to thrive in the highly dynamic habitat of sulfidic gas vents.

## Figures and Tables

**Figure 1 microorganisms-11-01366-f001:**
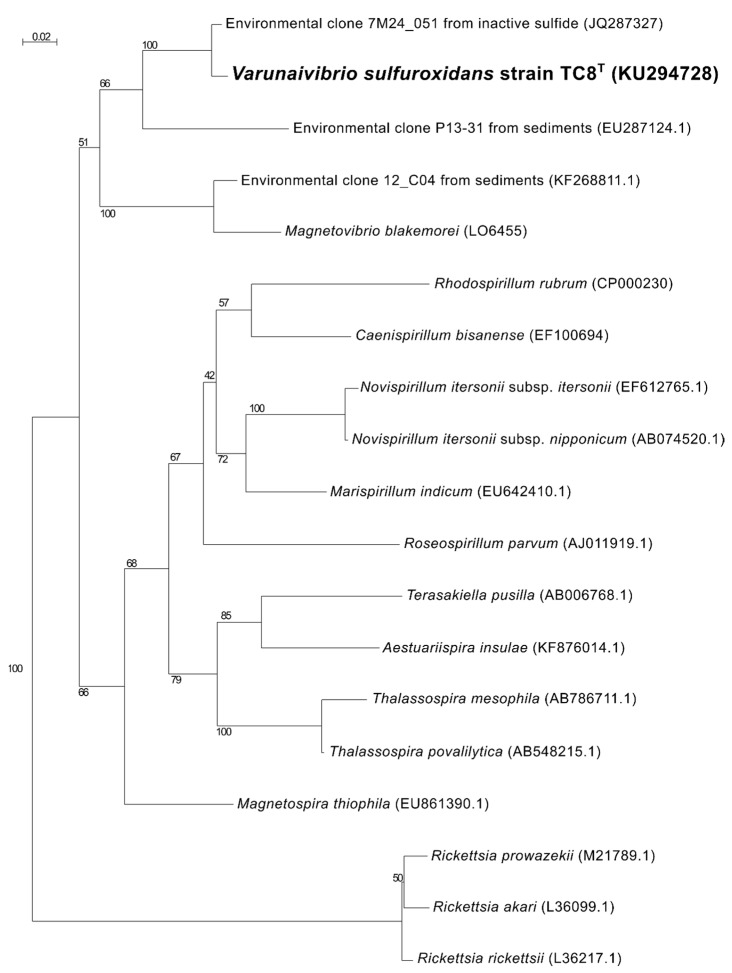
Maximum likelihood phylogenetic tree derived from 16S rRNA gene sequences showing the position of *Varunaivibrio sulfuroxidans* strain TC8^T^ (in bold) within the Alphaproteobacteria. Bootstrap values were based on 500 replicates and are shown at each node. Bar, 0.02% substitutions per position. Sequences belonging to the *Rickettsia* were used as the outgroup.

**Figure 2 microorganisms-11-01366-f002:**
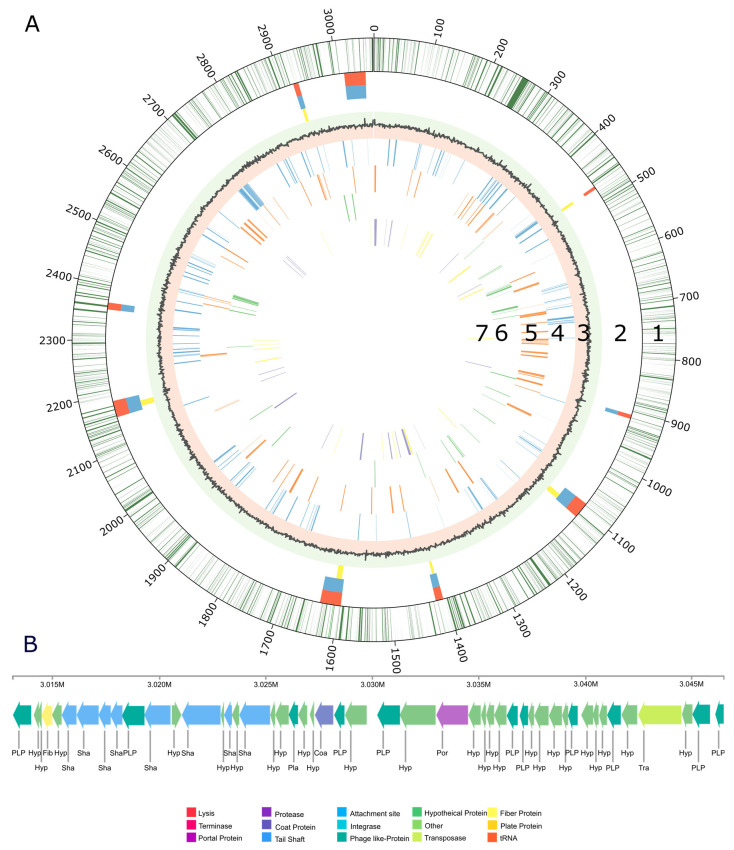
(**A**) Genome structure of *Varunaivibrio sulfuroxidans* strain TC8^T^. Features, starting with the outermost circle: 1. gene distribution within strain TC8^T^ genome; 2. predicted genomic islands in the genome of strain TC8^T^ by IslandPath-DIMOB (blue), SIGI-HMM (yellow); predicted by all tools within IslandViewer (orange); 3. GC skew; 4. genetic information processing (blue lines); 5. membrane transport (orange lines); 6. carbon metabolism: glycolysis/gluconeogenesis, TCA and CBB cycle (green lines); and 7. nitrogen metabolism (purple lines) sulfur metabolism (yellow lines). (**B**) Complete genome of the prophage of *Varunaivibrio sulfuroxidans* strain TC8^T^.

**Figure 3 microorganisms-11-01366-f003:**
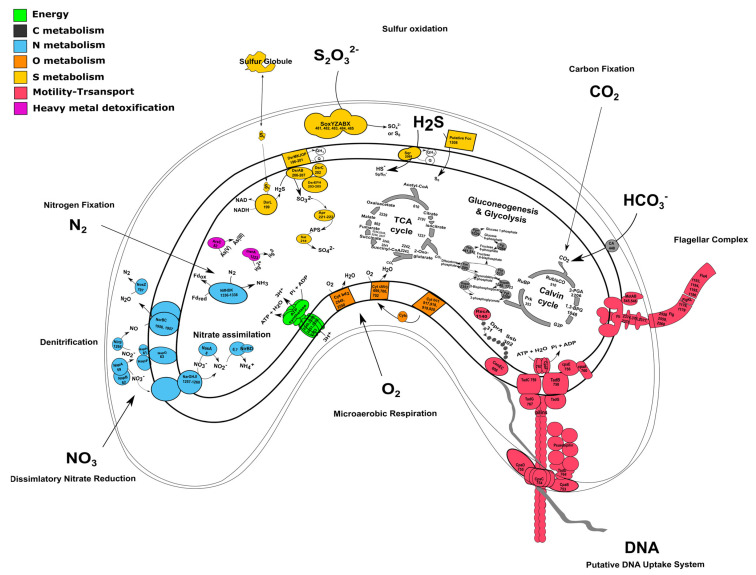
Genomic reconstruction of the central metabolism and DNA uptake system of *Varunaivibrio sulfuroxidans* strain TC8^T^. Enzymes are reported as in Appendix A. Abbreviations: *DENITRIFICATION*: NapABCFGHL: periplasmic nitrate reductase complex; NarGHJI: cytoplasmic nitrate reductase complex; NirS: nitrite reductase; NorBC: nitric oxide reductase; NosZ: nitrous oxide reductase. *ENERGY CONSERVATION*: ATP synthetase complex; *THIOSULFATE/SULFUR OXIDATION*: Sqr: sulfide:quinone oxidoreductase; Fcc: putative cytochrome c-sulfide dehydrogenase; SoxYZABX: thiosulfate oxidation; Dsr complex: sulfite reductase; AprAB: adenylylsulfate reductase; Sat: sulfate adenylyltransferase. *MICROAEROBIC RESPIRATION*: Cyt bd2 oxidase: cytochrome bd2; Cyt cbb3: cytochrome c oxidase; Cyt bc1: ubiquinol cytochrome c reductase. *NITRATE ASSIMILATION*: Nas. *NITROGEN FIXATION*: NifHDK: nitrogenase complex. *FLAGELLAR COMPLEX*: for simplicity, single unit names are not reported. *CITRIC ACID CYCLE*: citrate synthase (610); aconitate hydratase (2191); isocitrate dehydrogenase (1237); 2-oxoglutarate dehydrogenase complex (2242/2243); succinyl-CoA ligase (2240/2241); succinate dehydrogenase (2234/2235/2236/2237); fumarate hydratase (862); malate dehydrogenase (2239). *GLYCOLYSIS*: Eno: enolase; Pgm: phosphoglycerate mutase; Pgk: phosphoglycerate kinase; Gapdh: glyceraldehyde 3-phosphate dehydrogenase; Fba: fructose-bisphosphate aldolase; Tim: triosephosphate isomerase; Fbp: fructose-1,6-bisphosphatase; Pgi: phosphoglucose isomerase; Pgm: Phosphoglucomutase. *CALVIN–BENSON–BASSHAM CYCLE*: Prk: phosphoribulokinase. *CARBON CONCENTRATING MECHANISM (CCM)*: CA: carbonic anhydrase. *SECRETION SYSTEM* and *COMPETENCE*: TadBCDG and CpaABCDEF: Type IV secretion system complex and pilus assembly; ComEC: DNA internalization competence protein. *MERCURY REDUCTION*: MerA: mercuric ion reductase. *ARSENATE REDUCTION*: ArsC: arsenate reductase.

**Figure 4 microorganisms-11-01366-f004:**
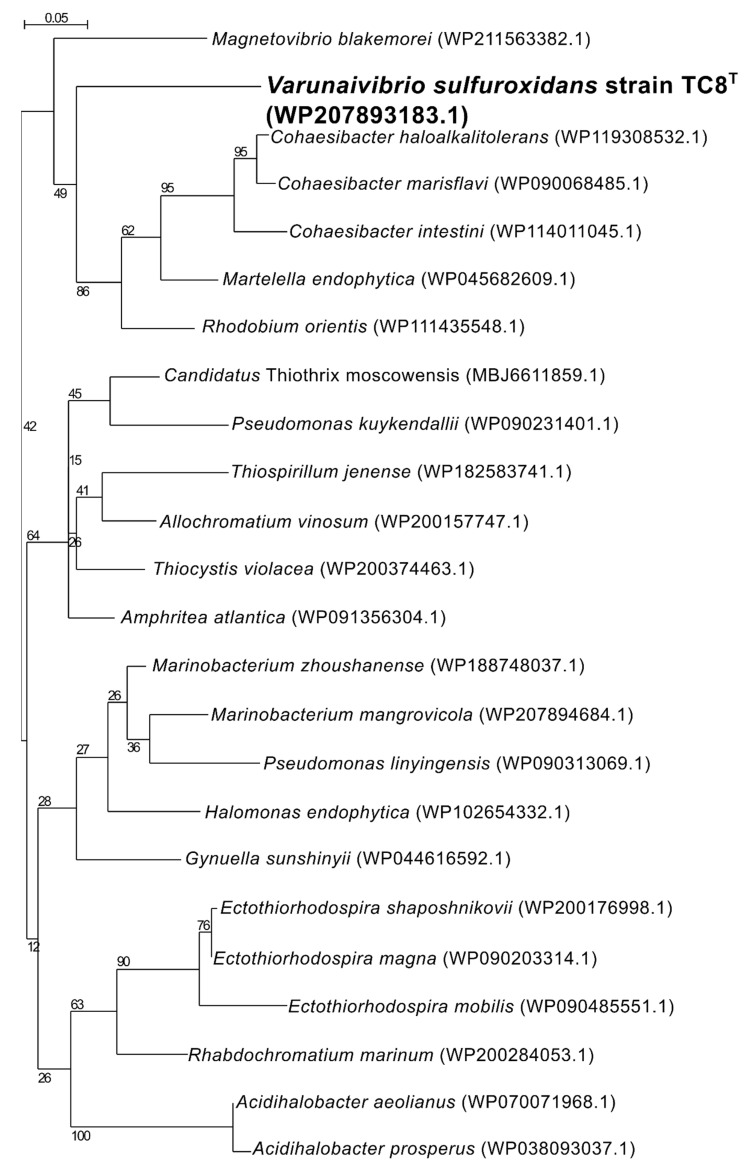
Maximum likelihood phylogenetic tree showing the position of the nitrogenase (NifH) of *Varunaivibrio sulfuroxidans* strain TC8^T^ (in bold). Bootstrap values based on 500 replications are shown at branch nodes. Bar, 0.05% substitutions per position.

**Figure 5 microorganisms-11-01366-f005:**
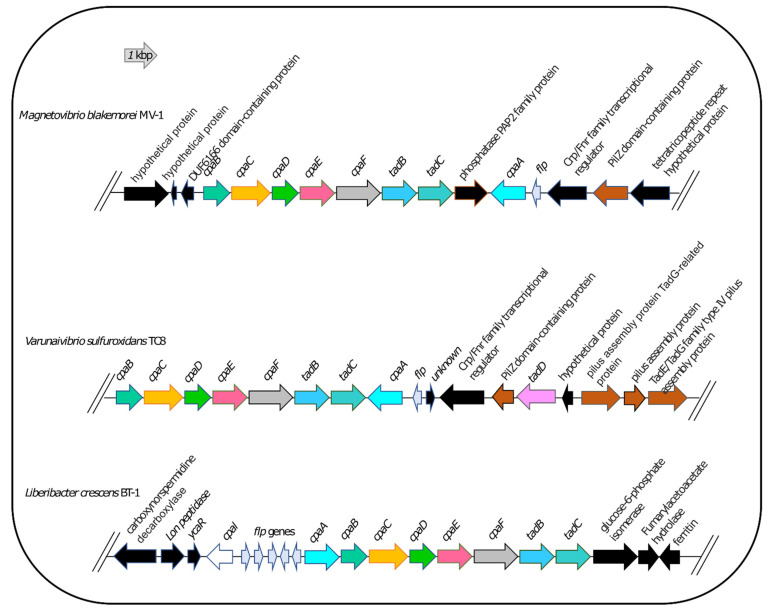
Organization of the *tad/cpa* gene cluster in *Magnetovibrio blakemorei* strain MV-1, *Varunaivibrio sulfuroxidans* strain TC8^T^ and *Liberibacter crescens* strain BT-1. Individual genes are drawn to scale.

**Figure 6 microorganisms-11-01366-f006:**
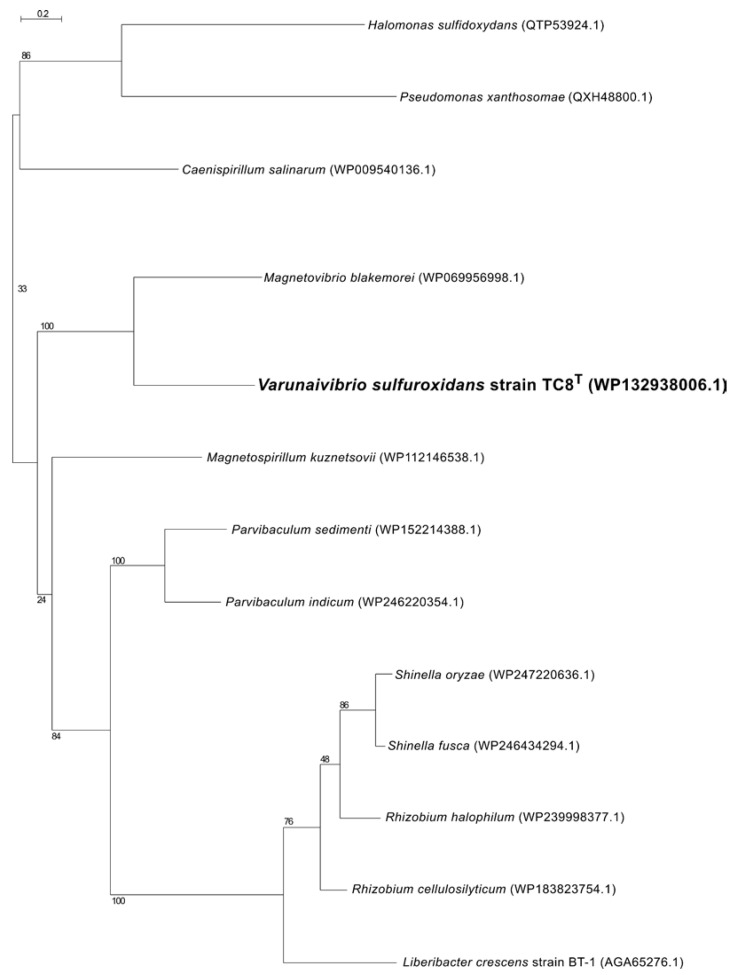
Maximum likelihood phylogenetic tree showing the position of the CpaC component of the secretion system of *Varunaivibrio sulfuroxidans* strain TC8^T^ (in bold). Bootstrap values based on 500 replications are shown at branch nodes. Bar, 0.2% substitutions per position.

**Figure 7 microorganisms-11-01366-f007:**
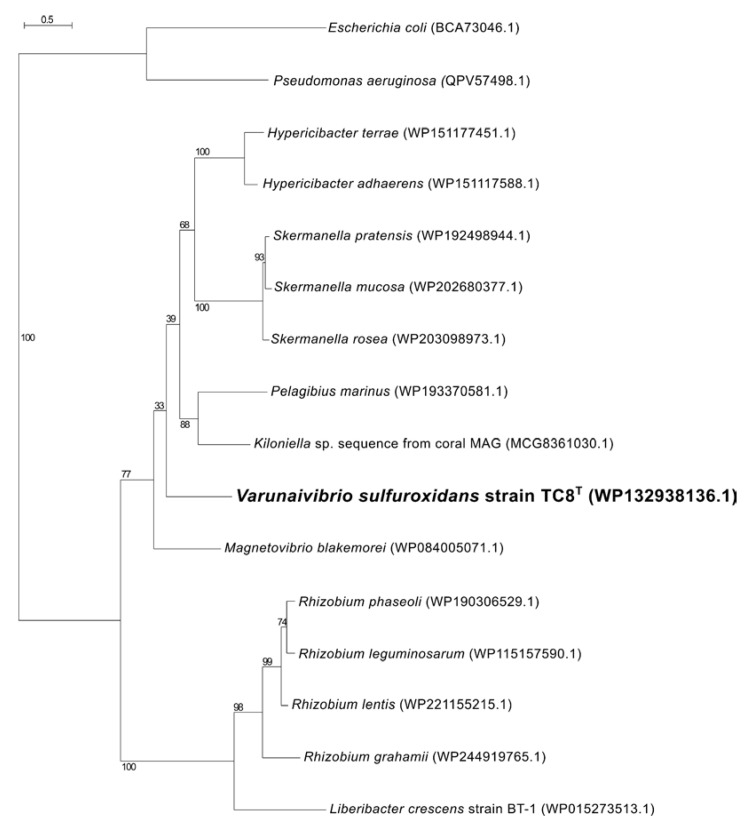
Maximum likelihood phylogenetic tree showing the position of the ComEC component of the competence apparatus of *Varunaivibrio sulfuroxidans* strain TC8^T^ (in bold). Bootstrap values based on 500 replications are shown at branch nodes. Bar, 0.5% substitutions per position.

**Table 1 microorganisms-11-01366-t001:** Genome statistics.

Attribute	Genome (Total)	
	Value	% Total ^1^
Size (bp)	3,066,297	100.00%
G+C content (bp)	1,821,569	59.41%
Coding region (bp)	2,702,087	88.12%
Total genes	2890	100.00%
RNA genes	61	2.11%
Protein-coding genes	2829	97.89%
Genes assigned to Pfam3	2450	84.78%
Genes assigned to COGs	2105	72.84%
Genes assigned in KEGG Orthology (KO)	1641	56.78%
Genes assigned to Subsystems	1568	54.25%
Genes coding signal peptides	160	5.54%
Genes coding transmembrane proteins	745	25.78%
CRISPRs	1	
Intact prophage	1	

^1^ The% total is based on either the size of the genome in base pairs or the total number of protein coding genes in the annotated genome.

**Table 2 microorganisms-11-01366-t002:** Number of genes associated with the 25 general COG functional categories.

Code	Value	% Total ^2^	Description
E	10.32%	239	Amino acid transport and metabolism
C	8.93%	207	Energy production and conversion
J	7.60%	176	Translation, ribosomal structure and biogenesis
P	7.51%	174	Inorganic ion transport and metabolism
R	6.78%	157	General function prediction only
H	5.91%	137	Coenzyme transport and metabolism
T	5.91%	137	Signal transduction mechanisms
O	5.74%	133	Posttranslational modification, protein turnover, chaperones
M	5.44%	126	Cell wall/membrane/envelope biogenesis
K	4.96%	115	Transcription
G	4.92%	114	Carbohydrate transport and metabolism
S	4.75%	110	Function unknown
I	3.84%	89	Lipid transport and metabolism
L	3.63%	84	Replication, recombination and repair
F	2.81%	65	Nucleotide transport and metabolism
N	2.68%	62	Cell motility
Q	1.90%	44	Secondary metabolites biosynthesis, transport and catabolism
V	1.77%	41	Defense mechanisms
U	1.42%	33	Intracellular trafficking, secretion, and vesicular transport
X	1.38%	32	Mobilome: prophages, transposons
D	1.25%	29	Cell cycle control, cell division, chromosome partitioning
W	0.47%	11	Extracellular structures
B	0.04%	1	Chromatin structure and dynamics
-	27.16%	785	Not in COG

^2^ The total is based on either the size of the genome in base pairs or the total number of protein coding genes in the annotated genome.

## Data Availability

The genome sequencing project of *V. sulfuroxidans* strain TC8^T^ is available under accession number CP119676.

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
