# Peer review of "The Genome of Varunaivibrio sulfuroxidans Strain TC8T, a Metabolically Versatile Alphaproteobacterium from the Tor Caldara Gas Vents in the Tyrrhenian Sea"

_microorganisms, 2023, doi:10.3390/microorganisms11061366_

Round 1

Reviewer 1 Report

Title: The genome of Varunaivibrio sulfuroxidans strain TC8T, a metabolically versatile alphaproteobacterium from the Tor Caldara gas vents in the Tyrrhenian Sea

Dear Editor and Authors,

Thank you for the opportunity to review this article which presents a very interesting discussion in microorganism characterization. Below I make my comments and observations on the manuscript. I hope my commentaries will improve the document.

Overview and general recommendation

In this article, the authors present the genome sequence of Varunaivibrio sulfuroxidans strain TC8 T, which is analyzed for its physiological and metabolic characteristics.

Major comments

I consider the paper is well written and the relevance is self-contained, since it is a full metabolic and physiological interpretation of the obtained sequences. The methods are fully explained, and they use very well known tools in their research.

My only comment is the quality of figures. All the images need to be improved, since they have a poor resolution and, in some cases, (Fig. 3) is almost impossible to read the information on them.

Minor comments

Along the text there are some typos, please correct them.

·        Line 80: “supplemented with100 ul of lysozyme (100 mg/ml) and in”

·        Line 82: (20mg/ml)

·        Line 109: The aminoacid sequences of NifH, ComEC

·        Lines 163 and 165, check the Font in TC8T

I consider the paper is well written, there are some typos, that can be easily corrected.  

Author Response

Reviewer 1 stated that this manuscript is interesting and well written. We corrected the typos mentioned by Reviewer 1.

Comment on the poor quality of the figures. The problem with the figure quality, a point expressed by both Reviewers, was outside of our control. We had submitted full-resolution figures as pdfs, but for some reason the figures were then inserted in the manuscript that was circulated to the Reviewers as low-resolution screenshots. As instructed by the editorial office, we have resubmitted all the figures as full-resolution png files, and we have been told that this will take care of the problem.

Reviewer 2 Report

microorganisms-2358649-v1_Review-round-1

General comments

This manuscript reports the whole genome analysis of Varunaivibrio sulfuroxidans strain TC8T. The manuscript is well-written; however, it lacks the reference to another genome of the same species (of the same strain from a different repository).  In addition, comparison with the genome of the important relative, Magnetovibrio blakemorei, is also needed to deepen the genomic analysis and discussion.

Along with the specific comments, my overall evaluation of the manuscript is “major revision” with particular reference to the genomes of:

Varunaivibrio sulfuroxidans DSM 101688 (= TC8T)

https://www.ncbi.nlm.nih.gov/genome/76350?genome_assembly_id=468404

and,

Magnetovibrio blakemorei MV-1T

https://www.ncbi.nlm.nih.gov/genome/46401?genome_assembly_id=283660

Specific Comments

L22-23 (in Abstract) “the bacterium is metabolically versatile and well-adapted to the dynamic environmental conditions of sulfidic gas vents.”

L398-399 (in Conclusions) “the metabolic versatility of this bacterium and corroborates its ability to thrive in the highly dynamic habitat of sulfidic gas vents.”

Metabolic versatility may be linked to changing environmental conditions. However, the authors state that “the (Tor Caldera) system is thought to be very stable over time in comparison to the more dynamic venting associated with active hydrothermal vents” (L39-41). Please give more consistent discussion about the metabolic versatility and adaptation.

L39 “… 600 ka ago”

This is a very minor comment, but I would point out that the geological term “ka” designates “kilo-years (thousand years) ago)” and contains the meaning of “ago” already. For most of the biological readers, I would suggest that the term “600 ka” should be spelled out as “600 thousand years ago” or “600,000 years ago”.

L56-57 “In particular, Magnetospira thiophila, Magnetovibrio blakemorei and Varunaivibrio sulfuroxidans …”

Please add brief profiles of the mentioned species other than Varunaivibrio sulfuroxidans.

Magnetospira thiophila MMS-1T, a marine magnetotactic bacterium was isolated from mud and water collected from School Street Marsh, Woods Hole, Massachusetts, on the north-east coast of USA (https://doi.org/10.1098/rspb.1993.0035).

Magnetovibrio blakemorei MV-1T, a magnetotactic bacterium (Alphaproteobacteria: Rhodospirillales: Thalassospiraceae) was isolated from a salt marsh pool at the Neponset River estuary near Boston, MA, USA (https://doi.org/10.1038/334518a0).

Both species show magnetotactism, and I just wondered if Varunaivibrio sulfuroxidans shows magnetotactism, too. This aspect should be mentioned in the sub-section “3.1. Overview of Varunaivibrio sulfuroxidans strain TC8T” and in the “4. Conclusions” section to improve the discussion of “metabolic versatility” (L398) that is possibly associated with magnetotactism as reviewed in a recent article (https://doi.org/10.1038/s41522-022-00304-0).

L64-65 “another being a Rhodovulum sp. from the coastal hydrothermal system of …”

Please do not miss the genome of Varunaivibrio sulfuroxidans DSM 101688 (= TC8T).

L145- “3.2. Genome Structure

If Varunaivibrio sulfuroxidans strain TC8T possesses no plasmids, please state so clearly somewhere in this sub-section.

Table 1 (and maybe Table 2)

Comparison of the studied 3.07 Mb genome of the Varunaivibrio sulfuroxidans TC8T with the 3.03 Mb genome of V. sulfuroxidans DSM 101688 (= TC8T) and the 3.64 Mb genome of Magnetovibrio blakemorei MV-1T is needed to make this manuscript more informative.

Varunaivibrio sulfuroxidans DSM 101688 (= TC8T):

https://www.ncbi.nlm.nih.gov/genome/76350?genome_assembly_id=468404

Magnetovibrio blakemorei MV-1T

https://www.ncbi.nlm.nih.gov/genome/46401?genome_assembly_id=283660

Figures 2 and 3

Details of this figure are mostly invisible. Please improve the visibility.

L349 “Figs. 2”

Is this a typo?.

L390-391 “… can take up exogenous DNA

If so, evidence/indication of horizontal gene transfer (HGT) may be found in their genomes. HGT-Finder is available online: https://doi.org/10.3390/toxins7104035

Further comments will be given after the major revision on these points.

Author Response

General comments

Reviewer 2 stated that this manuscript is well written, but pointed out that it does not reference two other genomes: that of Varunaivibrio sulfuroxidans strain TC8 and that of Magnetovibrio blakemorei. Hence Reviewer 2 recommended major revisions.

We respectfully point out that the other strain TC8 the reviewer refers to is in fact the same bacterium whose genome we are reporting on. This second genome entry is simply another sequencing project of the same strain, carried out simultaneously by the JGI. Hence the different repository – but it’s the same organism and the same genome, as there is only one Varunaivibrio sulfuroxidans strain TC8, the one reported in Patwardhan et al., 2016.

As for a comparison with the genome of Magnetovibrio blakemorei, we would like to point out that this manuscript has been conceived as a description of the genome of Varunaivibrio sulfuroxidans strain TC8, not as a comparative genomic manuscript between all the chemosynthetic marine alphaproteobacteria, which would include at least M. blakemorei and Magnetospira thiophila. That would be a completely different paper, which we had not intended to write, at least at this time.

Having said that, we’d like to point out that, whenever appropriate, we have indeed referred to the genome of M. blakemorei. Each time we discussed the main pathways encoded by the genome of strain TC8, we have made reference to the homologous enzymes for each of those pathways found in the genome of M. blakemorei, indicated the % identity, and we have included the enzymes of M. blakemorei in all the phylogenetic analyses (please see Figs. 4, 6 and 7).

We also carried out a Blast Dot Plot analysis of Varunaivibrio sulfuroxidans TC8 vs. Magnetovibrio blakemorei MV-1 (Fig. S1) as well as a comparative percent protein sequence identity between the two strains (Fig. S2). We did not include these analyses in the first draft of the manuscript, as we deemed them outside of the scope of this study. However, based on Reviewer 2 comments, we have now included these analyses in the supplemental material, including the side-by-side comparison of the genomes of V. sulfuroxidans TC8 and M. blakemorei MV-1, as requested by Reviewer 2 (Supplemental Table 2). However, an in-depth comparative genomic analysis of these strains would entail a completely different manuscript. It is not our intention to write that manuscript at this time.

Specific comments

Low resolution figures: see our explanation in the response to Reviewer 1.

Comment on the metabolic versatility and adaptation to the conditions of gas vents. When we state that Tor Caldara is a stable system compared to active submarine volcanoes, we refer to stability in geological time. The gas emissions are still very dynamic at the time scales experienced by the microorganisms, as the redox gradients between the reducing gases that are vented at the seafloor and the surrounding seaware are quite steep.

Comment on “600 ka: fixed

Suggestion to add a short profile of the close relatives to strain TC8: done

Comment on magnetotacticism in strain TC8: V. sulfuroxidans is not magnetotactic, formation of magnetosomes under strictly anaerobic conditions was not observed microscopically and its genome does not encode the mam genes which, in contrast, are present in the genome of M. blakemorei (Patwardhan et al., 2016 and Table S1). We added a sentence in the strain TC8 overview section stating this.

Comment on lack of plasmids in strain TC8: we added a sentence stating this in the genome structure section.

Reviewer 2 suggested to look for evidence of HGT using the online tool, HGT-finder. This was a great suggestion and we did the analysis. However, we found the HGT-finder tool to be exceedingly user unfriendly, hence we opted for alternative online tools. To detect possible horizontal gene transfer events, we used Alienness Web Server V.2.0 (Rancurel et al., 2017). Of all the genes analyzed, none was identified as a possible candidate for HGT. We added a sentence stating this in the conclusions.